# Reward Imputation with Sketching for Contextual Batched Bandits

**Xiao Zhang**[1,2], **Ninglu Shao**[1,2,*], **Zihua Si**[1,2,*], **Jun Xu**[1,2,†],
**Wenhan Wang**[3], **Hanjing Su**[3], **Ji-Rong Wen**[1,2]
[1] Gaoling School of Artificial Intelligence, Renmin University of China, Beijing, China
[2] Beijing Key Laboratory of Big Data Management and Analysis Methods, Beijing, China
[3] Tencent Inc., Shenzhen, China
{zhangx89, ninglu_shao, zihua_si, junxu, jrwen}@ruc.edu.cn
{justinsu, ezewang}@tencent.com

## Abstract

Contextual batched bandit (CBB) is a setting where a batch of rewards is observed from the environment at the end of each episode, but the rewards of the non-executed actions are unobserved, resulting in partial-information feedback. Existing approaches for CBB often ignore the rewards of the non-executed actions, leading to underutilization of feedback information. In this paper, we propose an efficient approach called Sketched Policy Updating with Imputed Rewards (SPUIR) that completes the unobserved rewards using sketching, which approximates the full-information feedbacks. We formulate reward imputation as an imputation regularized ridge regression problem that captures the feedback mechanisms of both executed and non-executed actions. To reduce time complexity, we solve the regression problem using randomized sketching. We prove that our approach achieves an instantaneous regret with controllable bias and smaller variance than approaches without reward imputation. Furthermore, our approach enjoys a sublinear regret bound against the optimal policy. We also present two extensions, a rate-scheduled version and a version for nonlinear rewards, making our approach more practical. Experimental results show that SPUIR outperforms state-of-the-art baselines on synthetic, public benchmark, and real-world datasets.

## 1 Introduction

Contextual bandits have gained significant popularity in solving sequential decision-making problems (Li et al., 2010; Lan and Baraniuk, 2016; Yom-Tov et al., 2017; Yang et al., 2021), where the agent continuously updates its decision-making policy fully online (i.e., at each step), considering the context and the received reward feedback to maximize cumulative rewards. In this paper, we address a more general setting called *contextual batched bandits* (CBB). In CBB, the decision process is divided into $N$ episodes, and within each episode, the agent interacts with the environment for a fixed number of $B$ steps. At the end of each episode, the agent collects reward feedbacks and contexts. Subsequently, the policy is updated using the collected data to guide the decision-making process in the subsequent episode. CBB offers a practical framework for real-world streaming applications (e.g., streaming recommendation (Zhang et al., 2021, 2022)). In the context of CBB settings, the batch size $B$, can be adjusted by the agent to achieve improved regret guarantees and meet the data throughput requirements based on the available computing resources (Zhou, 2023).

---

[*] Ninglu Shao and Zihua Si have made equal contributions to this paper.
[†] Corresponding author: Jun Xu.

37th Conference on Neural Information Processing Systems (NeurIPS 2023).

In bandit settings, it is common for the environment to only provide feedback on the rewards of executed actions to the agent, while concealing the rewards of non-executed actions. This type of limited feedback is referred to as *partial-information feedback* (also called "bandit feedback"). In CBB setting, existing approaches tend to overlook the potential rewards associated with non-executed actions. Instead, they address the challenge of partial-information feedback through an exploration-exploitation tradeoff in both the context space and reward space (Han et al., 2020; Zhang et al., 2020). However, CBB agents typically estimate and maintain reward models for the action-selection policy, thereby capturing some information about the potential rewards of non-executed actions. This additional reward structure information is available for policy updating in each episode but remains untapped by existing batched bandit approaches.

In the context of contextual bandit settings where the policy is updated online, several bias-correction approaches have been introduced to tackle the issue of partial-information feedback. Dimakopoulou et al. (2019) presented linear contextual bandits integrating the balancing approach from causal inference, which reweight the contexts and rewards by the inverse propensity scores. Chou et al. (2015) designed pseudo-reward algorithms for contextual bandits using upper confidence bound (UCB) strategy, which use a direct method to estimate the unobserved rewards. Kim and Paik (2019) focused on the correction of feedback bias for LASSO bandit with high-dimensional contexts, and applied the doubly-robust approach to the reward modification using average contexts. While these approaches have demonstrated effectiveness in contextual bandit settings, little attention has been given to addressing the under-utilization of partial-information feedback in CBB setting.

Theoretical and experimental analyses in Section 2 indicate that better performance of CBB is achievable if the rewards of the non-executed actions can be received. Motivated by these observations, we propose a novel reward imputation approach for the non-executed actions, which mimics the reward generation mechanisms of environments. We conclude our contributions as follows.

(1) To fully utilize feedback information in CBB, we formulate the reward imputation as a problem of imputation regularized ridge regression, where the policy can be updated efficiently using sketching.

(2) We prove that our reward imputation approach obtains a relative-error bound for sketching approximation, achieves an instantaneous regret with a controllable bias and a smaller variance than that without reward imputation, has a lower bound of the sketch size independently of the overall number of steps, enjoys a sublinear regret bound against the optimal policy, and reduces the time complexity from $O(Bd^2)$ to $O(cd^2)$ for each action in one episode, where $B$ denotes the batch size, $c$ the sketch size, and $d$ the dimension of the context space, satisfying $d < c < B$.

(3) We present two practical variants of our reward imputation approach, including the rate-scheduled version that sets the imputation rate without tuning, and the version for nonlinear rewards.

(4) We carried out extensive experiments on a synthetic dataset, the publicly available Criteo dataset, and a dataset from a commercial app to demonstrate our performance, empirically analyzed the influence of different parameters, and verified the correctness of the theoretical results.

**Related Work.** Recently, batched bandit has become an active research topic in statistics and learning theory including 2-armed bandit (Perchet et al., 2016), multi-armed bandit (Gao et al., 2019; Zhang et al., 2020; Wang and Cheng, 2020), and contextual bandit (Han et al., 2020; Ren and Zhou, 2020; Gu et al., 2021). Han et al. (2020) defined linear contextual bandits, and designed UCB-type algorithms for both stochastic and adversarial contexts, where true rewards of different actions have the same parameters. Zhang et al. (2020) provided methods for inference on data collected in batches using bandits, and introduced a batched least squares estimator for both multi-arm and contextual bandits. Recently, Esfandiari et al. (2021) proved refined regret upper bounds of batched bandits in stochastic and adversarial settings. There are several recent works that consider similar settings to CBB, e.g., episodic Markov decision process (Jin et al., 2018), LASSO bandits (Wang and Cheng, 2020). Sketching is another related technology that compresses a large matrix to a much smaller one by multiplying a (usually) random matrix while retaining certain properties (Woodruff, 2014), which has been used in online convex optimization (Calandriello et al., 2017; Zhang and Liao, 2019).

## 2 Problem Formulation and Analysis

Let $[x] = \{1, 2, \ldots, x\}$, $\mathcal{S} \subseteq \mathbb{R}^d$ be the context space whose dimension is $d$, $\mathcal{A} = \{A_j\}_{j \in [M]}$ the action space containing $M$ actions, $[\boldsymbol{A}; \boldsymbol{B}] = [\boldsymbol{A}^\intercal, \boldsymbol{B}^\intercal]^\intercal$, $\|\boldsymbol{A}\|_{\mathrm{F}}$, $\|\boldsymbol{A}\|_1$ $\|\boldsymbol{A}\|_2$ denote the Frobenius

**Protocol 1** Contextual Batched Bandit (CBB)

---

**INPUT:** Batch size $B$, number of episodes $N$, action space $\mathcal{A} = \{A_j\}_{j \in [M]}$, context space $\mathcal{S} \subseteq \mathbb{R}^d$
1: Initialize policy $p_0 \leftarrow \mathbf{1}/M$, sample data buffer $\mathcal{D}_1 = \{(\boldsymbol{s}_{0,b}, A_{I_{0,b}}, R_{0,b})\}_{b \in [B]}$ using initial policy $p_0$
2: **for** $n = 1$ **to** $N$ **do**
3:     Update the policy $p_n$ on $\mathcal{D}_n$
4:     **for** $b = 1$ **to** $B$ **do**
5:         Observe context $\boldsymbol{s}_{n,b}$ and choose $A_{I_{n,b}} \in \mathcal{A}$ following the updated policy $p_n(\boldsymbol{s}_{n,b})$
6:     **end for**
7:     $\mathcal{D}_{n+1} \leftarrow \{(\boldsymbol{s}_{n,b}, A_{I_{n,b}}, R_{n,b})\}_{b \in [B]}$, where $R_{n,b}$ denotes the reward of action $A_{I_{n,b}}$ on context $\boldsymbol{s}_{n,b}$
8: **end for**

---

norm, 1-norm, and spectral norm of a matrix $\boldsymbol{A}$, respectively, $\|\boldsymbol{a}\|_1$ and $\|\boldsymbol{a}\|_2$ be the $\ell_1$-norm and the $\ell_2$-norm of a vector $\boldsymbol{a}$, $\sigma_{\min}(\boldsymbol{A})$ and $\sigma_{\max}(\boldsymbol{A})$ denote the minimum and maximum of the of singular values of $\boldsymbol{A}$. In this paper, we focus on the setting of *Contextual Batched Bandits* (CBB) in Protocol 1, where the decision process is partitioned into $N$ episodes, and in each episode, CBB consists of two phases: (1) the *policy updating* approximates the optimal policy based on the received contexts and rewards; (2) the *online decision* chooses actions for execution following the updated and fixed policy $p$ for $B$ steps ($B$ is also called the *batch size*), and stores the context-action pairs and the observed rewards of the executed actions into a data buffer $\mathcal{D}$. The reward $R$ in CBB is a *partial-information feedback* where rewards are unobserved for the non-executed actions.

In contrast to the existing batched bandit setting (Han et al., 2020; Esfandiari et al., 2021), where the true reward feedbacks for all actions are controlled by the same parameter vector while the received contexts differ across actions at each step, we make the assumption that in CBB setting, the mechanism of true reward feedback varies across actions, while the received context is shared among actions. Formally, for any context $\boldsymbol{s}_i \in \mathcal{S} \subseteq \mathbb{R}^d$ and action $A \in \mathcal{A}$, we assume that the expectation of the true reward $R_{i,A}^{\text{true}}$ is determined by an unknown action-specific *reward parameter vector* $\boldsymbol{\theta}_A^* \in \mathbb{R}^d$: $\mathbb{E}[R_{i,A}^{\text{true}} \mid \boldsymbol{s}_i] = \langle \boldsymbol{\theta}_A^*, \boldsymbol{s}_i \rangle$ (the linear reward will be extended to the nonlinear case in Section 5). This setting for reward feedback matches many real-world applications, e.g., each action corresponds to a different category of candidate coupons in coupon recommendation, and the reward feedback mechanism of each category differs due to the different discount pricing strategies.

Next, we delve deeper into understanding the impact of unobserved feedbacks on the performance of policy updating in CBB setting. We first conducted an empirical comparison by applying the batch UCB policy (SBUCB) (Han et al., 2020) to environments under different proportions of received reward feedbacks. In particular, the agent under full-information feedback can receive all the rewards of the executed and non-executed actions, called *Full-Information CBB* (FI-CBB) setting. From Figure 1, we can observe that the partial-information feedbacks are damaging in terms of hurting the policy updating, and batched bandit policy can benefit from more reward feedbacks, where the performance of $80\%$ feedback is very close to that of FI-CBB.

Then, we prove the difference of instantaneous regrets between the CBB and FI-CBB settings in Theorem 1 (proof can be found in Appendix A).

**Theorem 1.** *For any action $A \in \mathcal{A}$ and context $\boldsymbol{s}_i \in \mathcal{S}$, let $\boldsymbol{\theta}_A^n$ be the reward parameter vector estimated by the batched UCB policy in the $n$-th episode. The upper bound of instantaneous regret (defined by $|\langle \boldsymbol{\theta}_A^n, \boldsymbol{s}_i \rangle - \langle \boldsymbol{\theta}_A^*, \boldsymbol{s}_i \rangle|$) in the FI-CBB setting is tighter than that in CBB setting (i.e., using the partial-information feedback).*

From Theorem 1, we can infer that utilizing partial-information feedbacks leads to a deterioration in the regret of the bandit policy. Ideally, the policy would be updated using full-information feedback. However, in CBB, full-information feedback is unavailable. Fortunately, in CBB, different reward parameter vectors are maintained and estimated separately for each action, and the potential reward

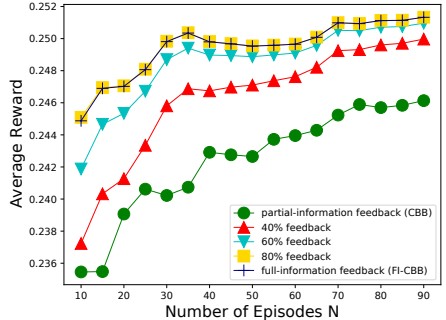

Figure 1: Average rewards of batch UCB policy (Han et al., 2020) under different proportions of received reward feedbacks, interacting with the synthetic environment in Section 6, where $x\%$ feedback means that $x\%$ of actions can receive their true rewards

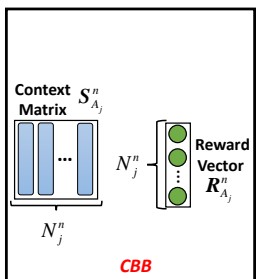 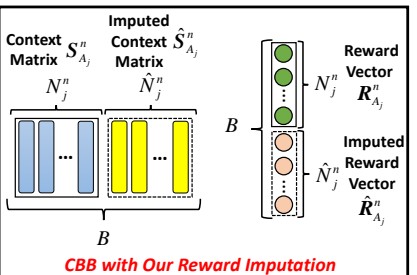 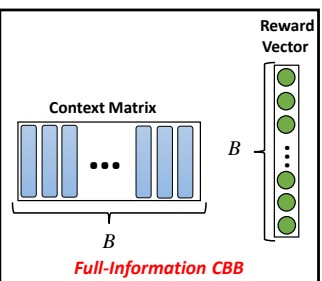

Figure 2: Comparison of the stored data corresponding to the action $A_j \in \mathcal{A} = \{A_j\}_{j \in [M]}$ in CBB, CBB with our reward imputation, and full-information CBB, in the $(n+1)$-th episode

structures of the non-executed actions have been captured to some extent. Therefore, why not utilize these maintained reward parameters to estimate the unknown rewards for the non-executed actions? In the following, we propose an efficient reward imputation approach that leverages this additional reward structure information to enhance the performance of the bandit policy.

# 3 Reward Imputation for Policy Updating

In this section, we present an efficient reward imputation approach tailored for policy updating in CBB setting.

**Formulation of Reward Imputation.** As shown in Figure 2, in contrast to CBB that ignores the contexts and rewards of the non-executed steps of each action, our reward imputation approach completes the missing values using the imputed contexts and rewards, approximating the full-information CBB setting. Specifically, at the end of the $(n+1)$-th episode, for each action $A_j \in \mathcal{A}, j \in [M]$, the context vectors and rewards received at the steps where the action $A_j$ is executed are observed, and are stored into a *context matrix* $\boldsymbol{S}_{A_j}^n \in \mathbb{R}^{N_j^n \times d}$ and a *reward vector* $\boldsymbol{R}_{A_j}^n \in \mathbb{R}^{N_j^n}$, respectively, where $N_j^n$ denotes the number of executed steps of $A_j$ in episode $n+1$. More importantly, at the steps (in episode $n+1$) where the action $A_j$ is NOT executed, the following imputations need to be performed for action $A_j$: (1) since the contexts are shared by all the actions, we directly store them into an *imputed context matrix* $\widehat{\boldsymbol{S}}_{A_j}^n \in \mathbb{R}^{\widehat{N}_j^n \times d}$, where $\widehat{N}_j^n$ denotes the number of non-executed steps of $A_j$ (i.e., $\widehat{N}_j^n = B - N_j^n$); (2) since the rewards of $A_j$ are unobserved at the non-executed steps, we estimate them using an *imputed reward vector*: for any $j \in [M]$,

$$\widehat{\boldsymbol{R}}_{A_j}^n = \{r_{n,1}(A_j), r_{n,2}(A_j), \ldots, r_{n,\widehat{N}_j^n}(A_j)\} \in \mathbb{R}^{\widehat{N}_j^n},$$

where $r_{n,b}(A_j) := \langle \bar{\boldsymbol{\theta}}_{A_j}^n, \boldsymbol{s}_{n,b} \rangle$ denotes the *imputed reward* parameterized by $\bar{\boldsymbol{\theta}}_{A_j}^n \in \mathbb{R}^d$ and $\boldsymbol{s}_{n,b}$ is the $b$-th row of $\widehat{\boldsymbol{S}}_{A_j}^n$.

Next, we introduce the updating process of the reward parameter vector $\bar{\boldsymbol{\theta}}_{A_j}^n$. We first concatenate the context and reward matrices from the previous episodes: $\boldsymbol{L}_{A_j}^n = [\boldsymbol{S}_{A_j}^0; \cdots; \boldsymbol{S}_{A_j}^n] \in \mathbb{R}^{L_j^n \times d}$, $\boldsymbol{T}_{A_j}^n = [\boldsymbol{R}_{A_j}^0; \cdots; \boldsymbol{R}_{A_j}^n] \in \mathbb{R}^{L_j^n}$, $L_j^n = \sum_{k=0}^n N_j^k$, $\widehat{\boldsymbol{L}}_{A_j}^n = [\widehat{\boldsymbol{S}}_{A_j}^0; \cdots; \widehat{\boldsymbol{S}}_{A_j}^n] \in \mathbb{R}^{\widehat{L}_j^n \times d}$, $\widehat{\boldsymbol{T}}_{A_j}^n = [\widehat{\boldsymbol{R}}_{A_j}^0; \cdots; \widehat{\boldsymbol{R}}_{A_j}^n] \in \mathbb{R}^{\widehat{L}_j^n}$, $\widehat{L}_j^n = \sum_{k=0}^n \widehat{N}_j^k$. Then, the *updated parameter vector* $\bar{\boldsymbol{\theta}}_{A_j}^{n+1}$ of the imputed reward for action $A_j$ can be obtained by solving the following *imputation regularized ridge regression*: for $n = 0, 1, \ldots, N-1$,

$$\bar{\boldsymbol{\theta}}_{A_j}^{n+1} = \arg\min_{\boldsymbol{\theta} \in \mathbb{R}^d} \underbrace{\left\| \boldsymbol{L}_{A_j}^n \boldsymbol{\theta} - \boldsymbol{T}_{A_j}^n \right\|_2^2}_{\text{Observed Term}} + \gamma \underbrace{\left\| \widehat{\boldsymbol{L}}_{A_j}^n \boldsymbol{\theta} - \widehat{\boldsymbol{T}}_{A_j}^n \right\|_2^2}_{\text{Imputation Term}} + \lambda \|\boldsymbol{\theta}\|_2^2, \tag{1}$$

where $\gamma \in [0, 1]$ is the *imputation rate* that controls the degree of reward imputation and measures a trade-off between bias and variance (Remark 1&2), $\lambda > 0$ is the regularization parameter. The discounted variant of the closed least squares solution of (1) is used for computing $\bar{\boldsymbol{\theta}}_{A_j}^{n+1}$:

$$\bar{\boldsymbol{\theta}}_{A_j}^{n+1} = \left( \boldsymbol{\Psi}_{A_j}^{n+1} \right)^{-1} \left( \boldsymbol{b}_{A_j}^{n+1} + \gamma \hat{\boldsymbol{b}}_{A_j}^{n+1} \right), \tag{2}$$

where $\boldsymbol{\Psi}_{A_j}^{n+1} := \lambda \boldsymbol{I}_d + \boldsymbol{\Phi}_{A_j}^{n+1} + \gamma \widehat{\boldsymbol{\Phi}}_{A_j}^{n+1}$, and

$$\boldsymbol{\Phi}_{A_j}^{n+1} = \boldsymbol{\Phi}_{A_j}^n + \boldsymbol{S}_{A_j}^{n\mathsf{T}} \boldsymbol{S}_{A_j}^n, \boldsymbol{b}_{A_j}^{n+1} = \boldsymbol{b}_{A_j}^n + \boldsymbol{S}_{A_j}^{n\mathsf{T}} \boldsymbol{R}_{A_j}^n, \tag{3}$$

$$\widehat{\boldsymbol{\Phi}}_{A_j}^{n+1} = \eta \widehat{\boldsymbol{\Phi}}_{A_j}^n + \widehat{\boldsymbol{S}}_{A_j}^{n\mathsf{T}} \widehat{\boldsymbol{S}}_{A_j}^n, \hat{\boldsymbol{b}}_{A_j}^{n+1} = \eta \hat{\boldsymbol{b}}_{A_j}^n + \widehat{\boldsymbol{S}}_{A_j}^{n\mathsf{T}} \widehat{\boldsymbol{R}}_{A_j}^n, \tag{4}$$

and $\eta \in (0,1)$ is the *discount parameter* that controls how fast the previous imputed rewards are forgotten, and can help guaranteeing the regret bound in Theorem 2.

**Efficient Reward Imputation using Sketching.** As shown in the first 4 columns in Table 1, the overall time complexity of the imputation for each action is $O(Bd^2)$ in each episode, where $B$ represents the batch size, and $d$ the dimensionality of the input. Thus, for all the $M$ actions in one episode, reward imputation increases the time complexity from $O(Bd^2)$ of the approach without imputation to $O(MBd^2)$. To address this issue, we design an efficient reward imputation approach using sketching, which reduces the time complexity of each action in one episode from $O(Bd^2)$ to $O(cd^2)$, where $c$ denotes the *sketch size* satisfying $d < c < B$ and $cd > B$. Specifically, in the $(n+1)$-th episode, the formulation in (1) can be approximated by a *sketched ridge regression* as:

$$\tilde{\boldsymbol{\theta}}_{A_j}^{n+1} = \underset{\boldsymbol{\theta} \in \mathbb{R}^d}{\arg \min} \left\| \boldsymbol{\Pi}_{A_j}^n \left( \boldsymbol{L}_{A_j}^n \boldsymbol{\theta} - \boldsymbol{T}_{A_j}^n \right) \right\|_2^2 + \gamma \left\| \widehat{\boldsymbol{\Pi}}_{A_j}^n \left( \widehat{\boldsymbol{L}}_{A_j}^n \boldsymbol{\theta} - \widehat{\boldsymbol{T}}_{A_j}^n \right) \right\|_2^2 + \lambda \|\boldsymbol{\theta}\|_2^2, \tag{5}$$

where $\tilde{\boldsymbol{\theta}}_A^{n+1}$ denotes the updated parameter vector of the imputed reward using sketching for action $A \in \mathcal{A}$, $\boldsymbol{C}_{A_j}^n \in \mathbb{R}^{c \times N_j^n}$ and $\widehat{\boldsymbol{C}}_{A_j}^n \in \mathbb{R}^{c \times \widehat{N}_j^n}$ are the *sketch submatrices* for the observed term and the imputation term, respectively, and the *sketch matrices* for the two terms can be represented as

$$\boldsymbol{\Pi}_{A_j}^n = \left[ \boldsymbol{C}_{A_j}^0, \boldsymbol{C}_{A_j}^1, \cdots, \boldsymbol{C}_{A_j}^n \right] \in \mathbb{R}^{c \times L_j^n}, \quad \widehat{\boldsymbol{\Pi}}_{A_j}^n = \left[ \widehat{\boldsymbol{C}}_{A_j}^0, \widehat{\boldsymbol{C}}_{A_j}^1, \cdots, \widehat{\boldsymbol{C}}_{A_j}^n \right] \in \mathbb{R}^{c \times \widehat{L}_j^n}.$$

We denote the sketches of the context matrix and the reward vector by $\boldsymbol{\Gamma}_{A_j}^n := \boldsymbol{C}_{A_j}^n \boldsymbol{S}_{A_j}^n \in \mathbb{R}^{c \times d}$ and $\boldsymbol{\Lambda}_{A_j}^n := \boldsymbol{C}_{A_j}^n \boldsymbol{R}_{A_j}^n \in \mathbb{R}^c$, the sketches of the imputed context matrix and the imputed reward vector by $\widehat{\boldsymbol{\Gamma}}_{A_j}^n := \widehat{\boldsymbol{C}}_{A_j}^n \widehat{\boldsymbol{S}}_{A_j}^n \in \mathbb{R}^{c \times d}$ and $\widehat{\boldsymbol{\Lambda}}_{A_j}^n := \widehat{\boldsymbol{C}}_{A_j}^n \widehat{\boldsymbol{R}}_{A_j}^n \in \mathbb{R}^c$. Similarly to (2), the discounted variant of the closed solution of (5) as follows:

$$\tilde{\boldsymbol{\theta}}_{A_j}^{n+1} = \left( \boldsymbol{W}_{A_j}^{n+1} \right)^{-1} \left( \boldsymbol{p}_{A_j}^{n+1} + \gamma \hat{\boldsymbol{p}}_{A_j}^{n+1} \right), \tag{6}$$

where $\eta \in (0,1)$ denotes the discount parameter, $\boldsymbol{W}_{A_j}^{n+1} := \lambda \boldsymbol{I}_d + \boldsymbol{G}_{A_j}^{n+1} + \gamma \widehat{\boldsymbol{G}}_{A_j}^{n+1}$, and

$$\boldsymbol{G}_{A_j}^{n+1} = \boldsymbol{G}_{A_j}^n + \boldsymbol{\Gamma}_{A_j}^{n\mathsf{T}} \boldsymbol{\Gamma}_{A_j}^n, \boldsymbol{p}_{A_j}^{n+1} = \boldsymbol{p}_{A_j}^n + \boldsymbol{\Gamma}_{A_j}^{n\mathsf{T}} \boldsymbol{\Lambda}_{A_j}^n, \tag{7}$$

$$\widehat{\boldsymbol{G}}_{A_j}^{n+1} = \eta \widehat{\boldsymbol{G}}_{A_j}^n + \widehat{\boldsymbol{\Gamma}}_{A_j}^{n\mathsf{T}} \widehat{\boldsymbol{\Gamma}}_{A_j}^n, \hat{\boldsymbol{p}}_{A_j}^{n+1} = \eta \hat{\boldsymbol{p}}_{A_j}^n + \widehat{\boldsymbol{\Gamma}}_{A_j}^{n\mathsf{T}} \widehat{\boldsymbol{\Lambda}}_{A_j}^n. \tag{8}$$

Using the parameter $\tilde{\boldsymbol{\theta}}_{A_j}^{n+1}$, we obtain the *sketched version of imputed reward* as $\tilde{r}_{n,b}(A_j) := \langle \tilde{\boldsymbol{\theta}}_{A_j}^n, \boldsymbol{s}_{n,b} \rangle$ at step $b \in [\widehat{N}_j^n]$. Finally, we specify that the sketch submatrices $\{\boldsymbol{C}_A^n\}_{A \in \mathcal{A}, n \in [N]}$ and $\{\widehat{\boldsymbol{C}}_{A_j}^n\}_{A \in \mathcal{A}, n \in [N]}$ are the block construction of Sparser Johnson-Lindenstrauss Transform (SJLT) (Kane and Nelson, 2014), where the sketch size $c$ is divisible by the number of blocks $D$[3]. As shown in the last 4 columns in Table 1, sketching reduces the time complexity of reward imputation from $O(MBd^2)$ to $O(Mcd^2)$ for all $M$ actions in one episode, where $c < B$. When $Mc \approx B$, the overall time complexity of our reward imputation using sketching is even comparable to that without reward imputation, i.e., a $O(Bd^2)$ time complexity.

**Updated Policy using Imputed Rewards.** Inspired by the UCB strategy (Li et al., 2010), the updated policy for online decision of the $(n+1)$-th episode can be formulated using the imputed rewards (parameterized by $\bar{\boldsymbol{\theta}}_A^{n+1}$ in (2)) or the sketched version of imputed rewards (parameterized by $\tilde{\boldsymbol{\theta}}_A^{n+1}$ in (6)). Specifically, for a new context $\boldsymbol{s}$,

• *origin policy* $\bar{p}_{n+1}$ selects the action as $A \leftarrow \arg\max_{A \in \mathcal{A}} \langle \bar{\boldsymbol{\theta}}_A^{n+1}, \boldsymbol{s} \rangle + \omega [\boldsymbol{s}^{\mathsf{T}} (\boldsymbol{\Psi}_A^{n+1})^{-1} \boldsymbol{s}]^{\frac{1}{2}}$,

• *sketched policy* $\tilde{p}_{n+1}$ selects the action as $A \leftarrow \arg\max_{A \in \mathcal{A}} \langle \tilde{\boldsymbol{\theta}}_A^{n+1}, \boldsymbol{s} \rangle + \alpha [\boldsymbol{s}^{\mathsf{T}} (\boldsymbol{W}_A^{n+1})^{-1} \boldsymbol{s}]^{\frac{1}{2}}$,

where $\omega \geq 0$ and $\alpha \geq 0$ are the regularization parameters in policy and their theoretical values are given in Theorem 4. We summarize the reward imputation using sketching and the sketched policy into Algorithm 2, called SPUIR. Similarly, we call the updating of the original policy that uses reward imputation without sketching, the Policy Updating with Imputed Rewards (PUIR).

[3]Since we set the number of blocks of SJLT as $D < d$, we omit $D$ in the complexity analysis.

Table 1: The time complexities of the original reward imputation in (1) (first 4 columns) and the reward imputation using sketching in (5) (last 4 columns) for action $A_j$ in the $(n+1)$-th episode, where $N_j^n$ ($\widehat{N}_j^n$) denotes the number of steps at which the action $A_j$ is executed (non-executed) in episode $n+1$, $\widehat{N}_j^n + N_j^n = B$, and the sketch size $c$ satisfying $d < c < B$ and $cd > B$ (MM: matrix multiplication; MI: matrix inversion; Overall: overall time complexity for action $A_j$ in one episode)

| Original reward imputation in (1) | | | | Reward imputation using sketching in (5) | | | |
|---|---|---|---|---|---|---|---|
| Item | Operation | Equation | Time | Item | Operation | Equation | Time |
| $\mathbf{\Phi}_{A_j}^{n+1}, \widehat{\mathbf{\Phi}}_{A_j}^{n+1}$ | MM | (3), (4) | $O(Bd^2)$ | $\boldsymbol{G}_{A_j}^{n+1}, \widehat{\boldsymbol{G}}_{A_j}^{n+1}$ | MM | (7), (8) | $O(cd^2)$ |
| $\boldsymbol{b}_{A_j}^{n+1}, \hat{\boldsymbol{b}}_{A_j}^{n+1}$ | MM | (3), (4) | $O(Bd)$ | $\boldsymbol{p}_{A_j}^{n+1}, \hat{\boldsymbol{p}}_{A_j}^{n+1}$ | MM | (7), (8) | $O(cd)$ |
| $(\mathbf{\Psi}_{A_j}^{n+1})^{-1}$ | MI | (2) | $O(d^3)$ | $(\boldsymbol{W}_{A_j}^{n+1})^{-1}$ | MI | (6) | $O(d^3)$ |
| | – | | | $\mathbf{\Gamma}_{A_j}^n, \mathbf{\Lambda}_{A_j}^n$ | Sketching | – | $O(N_j^n d)$ |
| | – | | | $\widehat{\mathbf{\Gamma}}_{A_j}^n, \widehat{\mathbf{\Lambda}}_{A_j}^n$ | Sketching | – | $O(\widehat{N}_j^n d)$ |
| Overall | – | – | $O(Bd^2)$ | Overall | – | – | $O(cd^2)$ |

---

**Algorithm 2** Sketched Policy Updating with Imputed Rewards (SPUIR) in the $(n+1)$-th episode

**INPUT:** Policy $\tilde{p}_n$, data buffer $\mathcal{D}_{n+1}$, $\mathcal{A} = \{A_j\}_{j\in[M]}$, $\alpha \geq 0$, $\eta \in (0,1)$, $\gamma \in [0,1]$, $\lambda > 0$, $\boldsymbol{W}_{A_j}^0 = \lambda \boldsymbol{I}_d$,
   $\boldsymbol{G}_{A_j}^0 = \widehat{\boldsymbol{G}}_{A_j}^0 = \boldsymbol{O}_d, \boldsymbol{p}_{A_j}^0 = \hat{\boldsymbol{p}}_{A_j}^0 = \boldsymbol{0}, \tilde{\boldsymbol{\theta}}_{A_j}^0 = \boldsymbol{0}, j \in [M]$, batch size $B$, sketch size $c$, number of block $D$
**OUTPUT:** Updated policy $\tilde{p}_{n+1}$
 1: For all $j \in [M]$, store context vectors and rewards corresponding to the steps at which the action $A_j$ is executed, into $\mathbf{\Gamma}_{A_j}^n \in \mathbb{R}^{N_j^n \times d}$ and $\mathbf{\Lambda}_{A_j}^n \in \mathbb{R}^{N_j^n}$
 2: For all $j \in [M]$, store context vectors corresponding to the steps at which the action $A_j$ is not executed into $\widehat{\mathbf{\Gamma}}_{A_j}^n \in \mathbb{R}^{\widehat{N}_j^n \times d}$, where $\widehat{N}_j^n \leftarrow B - N_j^n$
 3: $\tilde{r}_{n,b}(A_j) \leftarrow \langle \tilde{\boldsymbol{\theta}}_{A_j}^n, \boldsymbol{s}_{n,b} \rangle$, for all $A_j \in \mathcal{A}$ and $b \in [\widehat{N}_j^n]$, where $\boldsymbol{s}_{n,b}$ is the $b$-th row of $\widehat{\mathbf{\Gamma}}_{A_j}^n$
 4: Compute imputed reward vector $\widehat{\boldsymbol{R}}_{A_j}^n \leftarrow \{\tilde{r}_{n,1}(A_j), \dots, \tilde{r}_{n,\widehat{N}_j^n}(A_j)\} \in \mathbb{R}^{\widehat{N}_j^n}, \forall j \in [M]$
 5: **for all** action $A_j \in \mathcal{A}$ **do**
 6:    $\boldsymbol{G}_{A_j}^{n+1} \leftarrow \boldsymbol{G}_{A_j}^n + \mathbf{\Gamma}_{A_j}^{n\intercal}\mathbf{\Gamma}_{A_j}^n, \boldsymbol{p}_{A_j}^{n+1} \leftarrow \boldsymbol{p}_{A_j}^n + \mathbf{\Gamma}_{A_j}^{n\intercal}\mathbf{\Lambda}_{A_j}^n$ $\quad\{(7)\}$
       $\widehat{\boldsymbol{G}}_{A_j}^{n+1} \leftarrow \eta\widehat{\boldsymbol{G}}_{A_j}^n + \widehat{\mathbf{\Gamma}}_{A_j}^{n\intercal}\widehat{\mathbf{\Gamma}}_{A_j}^n, \hat{\boldsymbol{p}}_{A_j}^{n+1} \leftarrow \eta\hat{\boldsymbol{p}}_{A_j}^n + \widehat{\mathbf{\Gamma}}_{A_j}^{n\intercal}\widehat{\mathbf{\Lambda}}_{A_j}^n$ $\quad\{(8)\}$
 7:    $\boldsymbol{W}_{A_j}^{n+1} \leftarrow \lambda\boldsymbol{I}_d + \boldsymbol{G}_{A_j}^{n+1} + \gamma\widehat{\boldsymbol{G}}_{A_j}^{n+1}, \quad \tilde{\boldsymbol{\theta}}_{A_j}^{n+1} \leftarrow (\boldsymbol{W}_{A_j}^{n+1})^{-1}(\boldsymbol{p}_{A_j}^{n+1} + \gamma\hat{\boldsymbol{p}}_{A_j}^{n+1})$ $\quad\{(6)\}$
 8: **end for**
 9: $\tilde{p}_{n+1}(\boldsymbol{s})$ selects action $A \leftarrow \arg\max_{A\in\mathcal{A}} \langle \tilde{\boldsymbol{\theta}}_A^{n+1}, \boldsymbol{s} \rangle + \alpha[\boldsymbol{s}^\intercal (\boldsymbol{W}_A^{n+1})^{-1} \boldsymbol{s}]^{\frac{1}{2}}$ for a new context $\boldsymbol{s}$
10: **Return** $\{\tilde{\boldsymbol{\theta}}_A^{n+1}\}_{A\in\mathcal{A}}, \{(\boldsymbol{W}_A^{n+1})^{-1}\}_{A\in\mathcal{A}}$

---

## 4 Theoretical Analysis

We provide the instantaneous regret bound, prove the approximation error of sketching, and analyze the regret of SPUIR in CBB setting. The detailed proofs can be found in Appendix B. We first demonstrate the instantaneous regret bound of the original solution $\bar{\boldsymbol{\theta}}_A^n$ in (1).

**Theorem 2** (Instantaneous Regret Bound). *Let $\eta \in (0,1)$ be the discount parameter, $\gamma \in [0,1]$ the imputation rate. In the $n$-th episode, if the rewards $\{R_{n,b}\}_{b\in[B]}$ are independent[4] and bounded by $C_R$, then, for any $b \in [B], \forall A \in \mathcal{A}$, there exists $C_{\mathrm{Imp}} > 0$ such that, with probability at least $1 - \delta$,*

$$\left| \langle \bar{\boldsymbol{\theta}}_A^n, \boldsymbol{s}_{n,b} \rangle - \langle \boldsymbol{\theta}_A^*, \boldsymbol{s}_{n,b} \rangle \right| \leq \left[ \lambda \|\boldsymbol{\theta}_A^*\|_2 + \nu + \gamma^{\frac{1}{2}}\eta^{-\frac{1}{2}}C_{\mathrm{Imp}} \right] \left[ \boldsymbol{s}_{n,b}^\intercal (\mathbf{\Psi}_A^n)^{-1} \boldsymbol{s}_{n,b} \right]^{\frac{1}{2}}, \tag{9}$$

*where $\mathbf{\Psi}_A^n = \lambda\boldsymbol{I}_d + \mathbf{\Phi}_A^n + \gamma\widehat{\mathbf{\Phi}}_A^n, \nu = [2C_R^2 \log(2MB/\delta)]^{\frac{1}{2}}$. The first term on the right-hand side of (9) can be seen as the bias term for the reward imputation, while the second term is the variance term. The variance term of our algorithm is not larger than that without the reward imputation, i.e, for any $\boldsymbol{s} \in \mathbb{R}^d$,*

$$\left[ \boldsymbol{s}^\intercal (\mathbf{\Psi}_A^n)^{-1} \boldsymbol{s} \right]^{\frac{1}{2}} \leq \left[ \boldsymbol{s}^\intercal (\lambda\boldsymbol{I}_d + \mathbf{\Phi}_A^n)^{-1} \boldsymbol{s} \right]^{\frac{1}{2}}.$$

*Further, a larger imputation rate $\gamma$ leads to a smaller variance term $[\boldsymbol{s}^\intercal (\mathbf{\Psi}_A^n)^{-1} \boldsymbol{s}]^{\frac{1}{2}}$.*

---

[4]The assumption about conditional independence of the rewards is commonly used in the bandits literature, which can be ensured using a master technology as a theoretical construction (Auer, 2002; Chu et al., 2011).

**Remark 1** (Smaller Variance). *From Theorem 2, we can observe that our reward imputation achieves a smaller variance ($[\boldsymbol{s}_{n,b}^{\intercal}(\boldsymbol{\Psi}_A^n)^{-1}\boldsymbol{s}_{n,b}]^{\frac{1}{2}}$) than that without the reward imputation. By combining Theorem 2 and the proof of Theorem 1, we can obtain that the variance in instantaneous regret bound of SPUIR is in between the variances in full and partial information scenarios. Thus, reward imputation in SPUIR provides a promising way to use expert advice approaches for bandit problems.*

**Remark 2** (Controllable Bias). *Our reward imputation approach incurs a bias term $\gamma^{\frac{1}{2}}\eta^{-\frac{1}{2}}C_{\mathrm{Imp}}$ in addition to the two bias terms $\lambda\|\boldsymbol{\theta}_A^*\|_2$ and $\nu$ that exist in every existing UCB-based policy. But the additional bias term $\gamma^{\frac{1}{2}}\eta^{-\frac{1}{2}}C_{\mathrm{Imp}}$ is controllable due to the presence of imputation rate $\gamma$ that can help controlling the additional bias. Moreover, the term $C_{\mathrm{Imp}}$ in the additional bias can be replaced by a function $f_{\mathrm{Imp}}(n)$, and $f_{\mathrm{Imp}}(n)$ is monotonic decreasing w.r.t. number of episodes $n$ provided that the mild condition $\sqrt{\eta} = \Theta(d^{-1})$ holds (the definition and analysis about $f_{\mathrm{Imp}}$ can be found in Appendix B.1). Overall, the imputation rate $\gamma$ controls a trade-off between the bias term and the variance term, and we will design a rate-scheduled approach for automatically setting $\gamma$ in Section 5.*

**Remark 3** (Relationship with Existing Instantaneous Regrets). *According to the original definition in the context of online learning, the definition of instantaneous regret should be $\max_{A\in\mathcal{A}}\langle\boldsymbol{\theta}_A^*, \boldsymbol{s}_{n,b}\rangle - \langle\boldsymbol{\theta}_{A_{I_{n,b}}}^*, \boldsymbol{s}_{n,b}\rangle$. However, in the specific setting of contextual batched bandit (CBB) that is the focus of this paper, as derived in Appendix (second inequality of Eq. (48)), if we denote the upper bound of $|\langle\bar{\boldsymbol{\theta}}_A^n, \boldsymbol{s}_{n,b}\rangle - \langle\boldsymbol{\theta}_A^*, \boldsymbol{s}_{n,b}\rangle|$ as U, then 2U serves as an upper bound for instantaneous regret. Thus, in the context of CBB explored in this paper, we are interested in an upper bound for $|\langle\bar{\boldsymbol{\theta}}_A^n, s_{n,b}\rangle - \langle\boldsymbol{\theta}_A^*, s_{n,b}\rangle|$ and define it as the instantaneous regret bound.*

Although some approximation error bounds using SJLT have been proposed (Nelson and Nguyên, 2013; Kane and Nelson, 2014; Zhang and Liao, 2019), it is still unknown what is the lower bound of the sketch size while applying SJLT to the sketched ridge regression problem in our SPUIR. Next, we prove the approximation error as well as the lower bound of the sketch size in SPUIR. For convenience, we drop all the superscripts and subscripts in this result.

**Theorem 3** (Approximation Error Bound of Imputation using Sketching). *Denote the imputation regularized ridge regression function by $F(\boldsymbol{\theta})$ (defined in (1)) and the sketched ridge regression function by $F^{\mathrm{S}}(\boldsymbol{\theta})$ (defined in (5)) for reward imputation, whose solutions are $\bar{\boldsymbol{\theta}} = \arg\min_{\boldsymbol{\theta}\in\mathbb{R}^d} F(\boldsymbol{\theta})$ and $\tilde{\boldsymbol{\theta}} = \arg\min_{\boldsymbol{\theta}\in\mathbb{R}^d} F^{\mathrm{S}}(\boldsymbol{\theta})$. Let $\gamma \in [0,1]$ be the imputation rate, $\lambda > 0$ the regularization parameter, $\delta \in (0, 0.1]$, $\varepsilon \in (0,1)$, $\boldsymbol{L}_{\mathrm{all}} = [\boldsymbol{L}; \sqrt{\gamma}\widehat{\boldsymbol{L}}]$, and $\rho_\lambda = \|\boldsymbol{L}_{\mathrm{all}}\|_2^2/(\|\boldsymbol{L}_{\mathrm{all}}\|_2^2 + \lambda)$. If $\boldsymbol{\Pi}$ and $\widehat{\boldsymbol{\Pi}}$ are SJLT, assuming that $D = \Theta(\varepsilon^{-1}\log^3(d\delta^{-1}))$ and the sketch size $c = \Omega\left(d\operatorname{polylog}\left(d\delta^{-1}\right)/\varepsilon^2\right)$, with probability at least $1 - \delta$, the following results hold:*

$$F(\tilde{\boldsymbol{\theta}}) \leq (1 + \rho_\lambda\varepsilon)F(\bar{\boldsymbol{\theta}}), \quad \|\tilde{\boldsymbol{\theta}} - \bar{\boldsymbol{\theta}}\|_2 = O\left(\sqrt{\rho_\lambda}\varepsilon\right).$$

To measure the convergence of approximating the optimal policy in an online manner, we define the *regret* of SPUIR against the optimal policy as

$$\operatorname{Reg}(N, B) := \max_{A\in\mathcal{A}}\sum_{n\in[N], b\in[B]}[\langle\boldsymbol{\theta}_A^*, \boldsymbol{s}_{n,b}\rangle - \langle\boldsymbol{\theta}_{A_{I_{n,b}}}^*, \boldsymbol{s}_{n,b}\rangle],$$

where $I_{n,b}$ denotes the index of the executed action using the sketched policy $\tilde{p}_n$ (parameterized by $\{\tilde{\boldsymbol{\theta}}_A^n\}_{A\in\mathcal{A}}$) at step $b$ in the $n$-th episode. We final prove the regret bound of SPUIR.

**Theorem 4** (Regret Bound of SPUIR). *Let $T = BN$ be the overall number of steps, $\eta \in (0,1)$ be the discount parameter, $\gamma \in [0,1]$ the imputation rate, $\lambda > 0$ the regularization parameter, $C_{\boldsymbol{\theta}^*}^{\max} = \max_{A\in\mathcal{A}}\|\boldsymbol{\theta}_A^*\|_2$, $C_{\mathrm{Imp}}$ be the positive constant defined in Theorem 2. Assume that the conditional independence assumption in Theorem 2 holds and the upper bound of rewards is $C_R$, $M = O(\operatorname{poly}(d))$, $T \geq d^2$, $\nu = [2C_R^2\log(2MB/\delta_1)]^{\frac{1}{2}}$ with $\delta_1 \in (0,1)$, $\omega = \lambda C_{\boldsymbol{\theta}^*}^{\max} + \nu + \gamma^{\frac{1}{2}}\eta^{-\frac{1}{2}}C_{\mathrm{Imp}}, \alpha = \omega C_\alpha$, where $C_\alpha > 0$ which decreases with increase of $1/\varepsilon$ and $\varepsilon \in (0,1)$. Let $\delta_2 \in (0, 0.1]$, $\rho_\lambda < 1$ be the constant defined in Theorem 3, and $C_{\mathrm{reg}}$ be a positive constant that decreases with increase of $1/\varepsilon$. For the sketch matrices $\{\boldsymbol{\Pi}_A^n\}_{A\in\mathcal{A}, n\in[N]}$ and $\{\widehat{\boldsymbol{\Pi}}\}_{A\in\mathcal{A}, n\in[N]}$, assuming that the number of blocks in SJLT $D = \Theta(\varepsilon^{-1}\log^3(d\delta_2^{-1}))$, and the sketch size satisfying*

$$c = \Omega\left(d\operatorname{polylog}\left(d\delta_2^{-1}\right)/\varepsilon^2\right),$$

*then, for an arbitrary sequence of contexts* $\{s_{n,b}\}_{n \in [N], b \in [B]}$, *with probability at least* $1 - N(\delta_1 + \delta_2)$,

$$\text{Reg}(N, B) \leq 2\alpha C_{\text{reg}} \sqrt{10M} \log(T+1)(\sqrt{dT} + dB) + O\left(T\sqrt{\rho_\lambda \epsilon d}/B\right). \quad (10)$$

**Remark 4.** *Setting* $B = O(\sqrt{T/d})$, $\rho_\lambda \epsilon = 1/d$ *yields a sublinear regret bound of order* $\widetilde{O}(\sqrt{MdT})$[5] *provided that the sketch size* $c = \Omega(\rho_\lambda^2 d^3 \text{ polylog}(d\delta_2^{-1}))$. *We can observe that the lower bound of* $c$ *is independent of the overall number of steps* $T$, *and a theoretical value of the batch size is* $B = C_B \sqrt{T/d} = C_B^2 N/d$, *where setting* $C_B \approx 25$ *is a suitable choice that has been verified in the experiments in Section 6. In particular, when* $\rho_\lambda = O(1/d)$, *the sketch size of order* $c = \Omega(d \text{ polylog} d)$ *is sufficient to achieve a sublinear regret.*

From the theoretical results of regret, we can observe that our SPUIR admits several advantages: (a) The order of our regret bound (w.r.t. the overall number of steps) is not higher than those in the literature in the fully-online setting (Li et al., 2019; Dimakopoulou et al., 2019) that is a more simple setting than ours; (b) The first term in the regret bound (10) measures the performance of policy updating using imputed rewards (called "policy error"). From Theorem 2 and Remark 1&2, we obtain that, in each episode, our policy updating has a smaller variance than the policy without the reward imputation, and incurs a decreasing additional bias under mild conditions, leading to a tighter regret (i.e., smaller policy error) after some number of episodes. (c) The second term on the right-hand side of (10) is of order $O(T\sqrt{\rho_\lambda \epsilon d}/B)$, which is incurred by the sketching approximation using SJLT (called "sketching error"). This sketching error does not have any influence on the order of regret of SPUIR, which may even have a lower order with a suitable choice of $\rho_\lambda \varepsilon$, e.g., setting $\rho_\lambda \varepsilon = T^{-1/4} d^{-1}$ yields a sketching error of order $O(T^{3/8} d^{1/2})$ provided that $c = \Omega(\rho_\lambda^2 d^3 \text{ polylog}(d\delta_2^{-1})\sqrt{T})$.

At a fundamental level, the effectiveness of the proposed reward imputation can be attributed to the following two key factors:

(1) **Leveraging contextual bandit structure**: Traditional bandit methods only consider the structural assumptions for executed actions, leaving out non-executed ones. Our reward imputation approach incorporates a wide range of reward function structural assumptions, covering both executed and non-executed actions. By imputing missing rewards with observed data, we reduce the impact of missing data for a more accurate reward estimation.

(2) **Balancing exploration and exploitation**: Reward imputation's effectiveness arises from its impact on the exploration-exploitation trade-off. By incorporating imputed rewards, our proposed algorithms can make informed decisions even when observed rewards are incomplete. This enhances the agent's exploration strategy, helping it discover more valuable actions and reducing cumulative regret. Essentially, our reward computation approach approximates full-information feedback, mitigating the explore/exploit dilemma.

## 5 Extensions of Our Approach

To make the proposed reward imputation approach more feasible and practical, we tackle the following two research questions by designing variants of our approach following the theoretical results:

**RQ1 (Parameter Selection):** *Can we set the imputation rate* $\gamma$ *without tuning?*

**RQ2 (Nonlinear Reward):** *Can we apply the proposed reward imputation approach to the case where the expectation of true rewards is nonlinear?*

**Rate-Scheduled Approach.** For RQ1, we equip PUIR and SPUIR with a rate-scheduled approach, called PUIR-RS and SPUIR-RS, respectively. From Remark 1&2, a larger imputation rate $\gamma$ leads to a smaller variance while increasing the bias, while the bias term includes a monotonic decreasing function w.r.t. number of episodes under mild conditions. Therefore, we can gradually increase $\gamma$ with the number of episodes, avoiding the large bias at the beginning of reward imputation. Specifically, we set $\gamma = X\%$ for episodes from $(X - 10)\% \times N$ to $X\% \times N$, where $X \in [10, 100]$.

**Application to Nonlinear Rewards.** For RQ2, we provide nonlinear versions of reward imputation. We use linearization technologies of nonlinear rewards, rather than directly setting the rewards as nonlinear functions (Valko et al., 2013; Chatterji et al., 2019), avoiding the linear regret or curse of

---

[5]We use the notation of $\widetilde{O}$ to suppress logarithmic factors in the overall number of steps $T$.

Table 2: Performance comparison of coupon recommendation on `commercial product`

| Algorithm | CVR (mean ± std) | CTCVR (mean ± std) | Time (sec., mean ± std) |
|---|---|---|---|
| DFM-S | $0.8656 \pm 0.0473$ | $0.3317 \pm 0.0218$ | $302.3140 \pm 8.3045$ |
| SBUCB | $0.8569 \pm 0.0037$ | $0.4277 \pm 0.0084$ | $43.5435 \pm 0.3659$ |
| BEXP3 | $0.4846 \pm 0.0205$ | $0.2425 \pm 0.0116$ | $53.5001 \pm 0.9220$ |
| BEXP3-IPW | $0.4862 \pm 0.0187$ | $0.2436 \pm 0.0113$ | $56.0101 \pm 1.4142$ |
| BLTS-B | $0.8663 \pm 0.0178$ | $0.4285 \pm 0.0157$ | $218.2109 \pm 1.8198$ |
| PUIR | $0.8807 \pm 0.0053$ | $0.4411 \pm 0.0029$ | $184.3575 \pm 2.2346$ |
| SPUIR | $0.8770 \pm 0.0059$ | $0.4397 \pm 0.0032$ | $81.5753 \pm 1.5879$ |
| PUIR-RS | $0.8763 \pm 0.0056$ | $0.4389 \pm 0.0030$ | $180.4999 \pm 1.7763$ |
| SPUIR-RS | $0.8758 \pm 0.0058$ | $0.4391 \pm 0.0031$ | $80.8003 \pm 2.9030$ |

kernelization. Specifically, instead of using the linear imputed reward $\tilde{r}_{n,b}(A_j) := \langle \tilde{\boldsymbol{\theta}}^n_{A_j}, \boldsymbol{s}_{n,b} \rangle$, we use the following linearized nonlinear imputed rewards, denotes by $\mathcal{T}_{n,b}(\boldsymbol{\theta}, A)$:

**(1) SPUIR-Exp.** We assume that the expected reward is an exponential function as $G_{\mathrm{E}}(\boldsymbol{\theta}, \boldsymbol{s}) = \exp\left(\boldsymbol{\theta}^\intercal \boldsymbol{s}\right)$. Then $\mathcal{T}_{n,b}(\boldsymbol{\theta}, A) = \langle \boldsymbol{\theta}, \nabla_{\boldsymbol{\theta}} G_{\mathrm{E}}(\boldsymbol{\theta}, \boldsymbol{s}_{n,b}) \rangle$, where $\nabla_{\boldsymbol{\theta}} G_{\mathrm{E}}(\boldsymbol{\theta}, \boldsymbol{s}_{n,b}) = \exp\left(\boldsymbol{\theta}^\intercal \boldsymbol{s}_{n,b}\right) \boldsymbol{s}_{n,b}$.

**(2) SPUIR-Poly.** When the expected reward is a polynomial function as $G_{\mathrm{P}}(\boldsymbol{\theta}, \boldsymbol{s}) = \left(\boldsymbol{\theta}^\intercal \boldsymbol{s}\right)^2$. Then $\mathcal{T}_{n,b}(\boldsymbol{\theta}, A) = \langle \boldsymbol{\theta}, \nabla_{\boldsymbol{\theta}} G_{\mathrm{P}}(\boldsymbol{\theta}, \boldsymbol{s}_{n,b}) \rangle$, where $\nabla_{\boldsymbol{\theta}} G_{\mathrm{P}}(\boldsymbol{\theta}, \boldsymbol{s}_{n,b}) = 2 \left(\boldsymbol{\theta}^\intercal \boldsymbol{s}_{n,b}\right) \boldsymbol{s}_{n,b}$.

**(3) SPUIR-Kernel.** Consider that the underlying expected reward in a Gaussian reproducing kernel Hilbert space (RKHS). We use $\mathcal{T}_{n,b}(\boldsymbol{\theta}, A) = \langle \boldsymbol{\theta}, \phi(\boldsymbol{s}_{n,b}) \rangle$ in random feature space, where the random feature mapping $\phi$ can be explicitly computed.

For SPUIR-Exp and SPUIR-Poly, combining the linearization of convex functions (Shalev-Shwartz, 2011) with Theorem 4 yields the regret bounds of the same order. For SPUIR-Kernel, using the approximation error of random features (Rahimi and Recht, 2008), we can also obtain that, SPUIR-Kernel has the same regret bound as SPUIR under mild conditions (see proofs in Appendix B).

## 6  Experiments

We empirically evaluated the performance of our algorithms on 3 datasets: the synthetic dataset, publicly available Criteo dataset[6] (`Criteo-recent`, `Criteo-all`), and dataset collected from Tencent's WeChat app for coupon recommendation (`commercial product`).

**Experimental Settings.** We compared our algorithms with: Sequential Batch UCB (SBUCB) (Han et al., 2020), Batched linear EXP3 (BEXP3) (Neu and Olkhovskaya, 2020), Batched linear EXP3 using Inverse Propensity Weighting (BEXP3-IPW) (Bistritz et al., 2019), Batched Balanced Linear Thompson Sampling (BLTS-B) (Dimakopoulou et al., 2019), and Sequential version of Delayed Feedback Model (DFM-S) (Chapelle, 2014). We applied the algorithms to CBB setting and implemented on Intel(R) Xeon(R) Silver 4114 CPU@2.20GHz, and repeated the experiments 20 times. We tested the performance of algorithms in streaming recommendation scenarios, where the reward is represented by a linear combination of the click and conversion behaviors of users. According to Remark 4, we set the batch size as $B = C_B^2 N/d$, the constant $C_B \approx 25$, and the sketch size $c = 150$ on all the datasets. The average reward was used to evaluate the accuracy of algorithms.

**Performance Evaluation.** Figure 3(a)–(c) reports the average reward of SPUIR with its variants and the baselines. We observed that SPUIR and its variants achieved higher average rewards, demonstrating the effectiveness of our reward imputation. Moreover, SPUIR and its rate-scheduled version SPUIR-RS had similar performances compared with the origin PUIR, which indicates the practical effectiveness of our variants and verifies the correctness of the theoretical analyses. The results on `commercial product` in Table 2 indicate that SPUIR outperformed the second-best baseline with the improvements of 1.07% CVR (conversion rate) and 1.12% CTCVR (post-view

---

[6]https://labs.criteo.com/2013/12/conversion-logs-dataset/

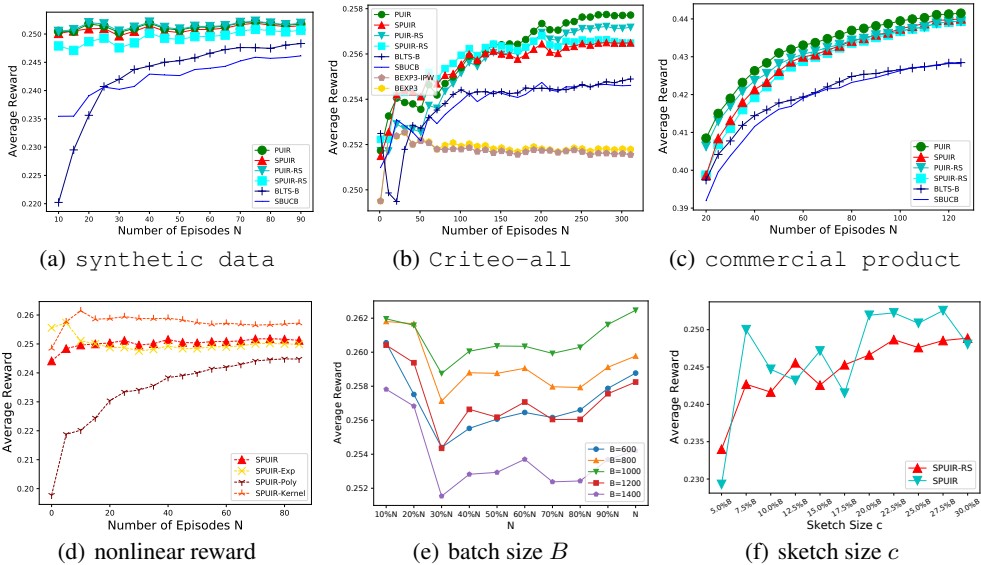

Figure 3: (a), (b), (c): Average rewards of the compared algorithms, the proposed SPUIR and its variants on synthetic dataset, Criteo dataset, and the real commercial product data, where we omitted the curves of algorithms whose average rewards are $5\%$ lower than the highest reward; (d): SPUIR and its three nonlinear variants on synthetic dataset; (e): SPUIR with different batch sizes on `Criteo-recent`; (f): SPUIR and SPUIR-RS with different sketch sizes on synthetic dataset

click-through&conversion rate). Besides, our reward imputation approaches were more efficient than DFM-S, BLTS-B. The variants using sketching of our algorithms (SPUIR, SPUIR-RS) significantly reduced the time costs of reward imputation, and took less than twice as long to run compared to the baselines without reward imputation (SBUCB, BEXP3, BEXP3-IPW). Figure 3(d) illustrates performances of SPUIR and its nonlinear variants, where SPUIR-Kernel achieved the highest rewards indicating the effectiveness of the nonlinear generalization of our approach. For different decision tasks, a suitable nonlinear reward model needs to be selected for better performances.

**Parameter Influence.** From the regret bound (10), we can observe that a larger batch size $B$ results in a larger first term (of order $O(B)$, called policy error) but a smaller second term (of order $O(1/B)$, called sketching error), indicating that a suitable batch size $B$ needs to be set. This conclusion was empirically verified in Figure 3(e), where $B = 1,000$ ($C_B = 25$) yields better empirical performance in terms of the average reward. Similar phenomenon can also be observed on Criteo dataset and `commercial product`. All of the results verified the theoretical results in Remark 4: $B = C_B\sqrt{T/d} = C_B^2 N/d$ is a suitable choice while setting $C_B \approx 25$. From the results in Figure 3(f) we observe that, for our SPUIR and SPUIR-RS, the performances significantly increased when the sketch size $c$ reached $10\%B$ ($\approx d\log d$), which demonstrates the conclusion in Remark 4 that only the sketch size of order $c = \Omega(d\,\mathrm{polylog}d)$ is needed for satisfactory performance.

# 7 Conclusion

This paper presents a computationally efficient reward imputation approach for contextual batched bandits that addresses the challenge of partial-information feedback in real-world applications. The proposed approach mimics the reward generation mechanism of the environment, approximating full-information feedback. It reduces time complexity using sketching, achieves a relative-error bound for approximation, and exhibits regret with controllable bias and reduced variance. The theoretical formulation and algorithmic implementation may provide an efficient reward imputation scheme for online learning under limited feedback.

## Acknowledgments and Disclosure of Funding

This work was funded by the National Key R&D Program of China (2022ZD0114802), the National Natural Science Foundation of China (62376275, U2001212, 62377044), Intelligent Social Governance Interdisciplinary Platform, Major Innovation & Planning Interdisciplinary Platform for the "Double-First Class" Initiative, Renmin University of China. Supported by fund for building world-class universities (disciplines) of Renmin University of China. Supported by Public Computing Cloud, Renmin University of China. Supported by the Fundamental Research Funds for the Central Universities, and the Research Funds of Renmin University of China (23XNKJ13).

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
