# OpenReview forum: "Reward Imputation with Sketching for Contextual Batched Bandits"
_NeurIPS.cc/2023/Conference — NeurIPS 2023 poster_

### Official Review · Reviewer_et1K · 2023-06-13

**Soundness:** 3 good
**Presentation:** 2 fair
**Contribution:** 3 good
**Rating:** 6
**Confidence:** 3

**Summary:**

This paper focuses on contextual batched bandit (CBB) problems, where only partial information (the rewards of executed actions) is available. The authors proposed an efficient reward imputation approach in CBB, which completes the unobserved reward, and the new objective can be solved by the imputation regularized ridge regression. They further proposed sketching to reduce time complexity. Theoretical guarantees are established, showing that this approach has a sublinear regret bound against the optimal policy.  Experiments are conducted, showing its superiority over baselines.

**Strengths:**

The paper is well-structured. The proposed reward imputation for the non-executed actions is innovative, and this idea seems widely applicable in many scenarios. Proof-of-concept experimental results support their theoretical claims.

**Weaknesses:**

A high-level idea of reward imputation needs to be provided, especially regarding why reward imputation is effective statistically. At a high level, it seems counter-intuitive since the imputed rewards are derived from the observed rewards, so it should not yield any statistical improvement. If there is indeed no improvement, the position of this work would be in question. However, the authors did achieve improvement, so it would be valuable for them to offer a high-level explanation regarding why reward imputation works in this context.

In addition, the absence of definitions for certain notations in the paper somewhat makes it challenging to evaluate certain results. I apologize if I may have overlooked these definitions in the paper. I have listed these issues in the "Questions" section below.

**Questions:**

- Is the batch size $B$ predetermined by the problem setup, or can it be manually adjusted by the learner? In the analysis conducted by the authors (e.g., Remark 3), they define $B$ to be on the order of $O(\sqrt{T})$, indicating that it is likely the latter case. In that context, what are the theoretical advantages of this algorithm when setting $B=1$ compared to standard contextual bandit algorithms? Can we simply apply CB algorithms to this particular setting with $B=1$ then?

- It looks like eq. (2) is a solution to eq. (1), which makes me wonder where the discount parameter $\eta$ in line 156 comes from. Is $\eta$ a predetermined hyperparameter or an arbitrary parameter? If it is the former, I did not find it in the problem setup; if it is the latter, it seems so weird as it suggests that any $\eta$ would make eq. (2) a solution to eq. (1)?

- Why is Theorem 2 referred to as the "instantaneous regret bound"? Shouldn't the instantaneous regret involve $\max\_a\langle\theta^*\_A,s\_{n,b}\rangle$? This theorem seems to be more of a bound on estimation error.

**Limitations:**

No potential negative societal impact.

---

> ### Author Rebuttal · Authors · 2023-08-09
>
> # Response to Reviewer et1K
>
> Thank you for your reviews and suggestions.
>
> **1. Response to Weakness**
>
> We appreciate the reviewer's insightful suggestion regarding the high-level idea. At first glance, the concept of reward imputation might appear counter-intuitive, as imputed rewards are derived from observed rewards, seemingly not providing any statistical advantage. However, in the context of our work, reward imputation proves effective due to the several key factors.
>
> Reward imputation leverages the inherent structure within contextual bandit problems. Reward imputation capitalizes on the intrinsic structure inherent to contextual bandit problems. More specifically, contextual bandit problems often assume that the expected rewards for each arm are functions of the context (e.g., linear or nonlinear functions as explored in this paper). Traditionally, the structural assumptions of these reward functions are solely employed on executed actions within each interaction round, neglecting non-executed actions. The pivotal aspect introduced by our reward imputation approach lies precisely in incorporating a comprehensive range of reward function structural assumptions, encompassing both executed and non-executed actions. By utilizing observed rewards to impute unobserved or missing rewards, we create a more comprehensive dataset that captures underlying patterns and relationships. The introduction of imputed rewards fills information gaps and mitigates the influence of missing data, ultimately leading to a more accurate estimation of the true reward distribution.
>
> Moreover, the effectiveness of reward imputation stems from the interplay between exploration and exploitation. By incorporating imputed rewards, our proposed algorithms make informed decisions even in situations where observed rewards might be incomplete. This improves the agent's exploration strategy, enabling it to discover more valuable actions, which in turn contributes to the cumulative regret reduction. In essence, our reward computation approach approximates a scenario resembling full-information feedback, which mitigates the explore/exploit dilemma, introduces a lower variance, and introduces a controlled additional bias component within the regret. The extra information influencing policy towards exploitation and away from exploration emanates from the estimated reward structures of non-executed actions maintained within each episode. Our reward imputation approach effectively captures this additional information, underscoring its effectiveness and efficiency.
>
> In summary, the key to understanding why reward imputation works in this context lies in the fact that it enhances the agent's decision-making process by effectively utilizing available data and reducing the impact of missing rewards. This, in turn, strengthens the exploration-exploitation trade-off, leading to the observed improvements in regret reduction.
>
> **2. Response to Question 1: Batch Size**
>
> In practical online environments, bandit polices involve selecting an arm to execute in each round of interaction, while the batch size merely determines how often the bandit policy is updated. This update frequency can certainly be manually adjusted by the learner based on computational constraints, which is the specific focus of this paper.
>
> The reward imputation approach we propose is applicable to scenarios with any batch size, including the case where $B=1$. Remark 3 analyzes the regret bound of SPUIR, which employs sketching techniques to accelerate the processing speed of each data batch. The theoretical value $B=O(\sqrt{T/d})$ strikes the optimal balance between sketching approximation error and regret. Even in the setting of standard contextual bandit algorithms (i.e., $B=1$), the proposed reward imputation approach still holds advantages in reducing regret. Specifically, as Theorem 2 indicates, even when $B=1$, our reward imputation approach continues to yield smaller variance and a controllable bias, ensuring its effectiveness.
>
> **3. Response to Question 2: Discount Parameter**
>
> When the discount parameter $\eta = 1$, Eq.(2) represents the closed least squares solution of Eq.(1). As mentioned in line 155, we refer to Eq.(2) as the "discounted variant of the closed least squares solution," which introduces decay to the historical context matrix and reward vector on top of the original closed least squares solution. We observe that the discount parameter allows for control over the rate at which previous imputed rewards (which may be inaccurately estimated) are forgotten. This control can contribute to ensuring the regret bound as presented in Theorem 2. The theoretical value of $\eta$ is discussed in Remark 2. Since the formulation of the regression problem corresponding to this discounted variant of the closed least squares solution becomes intricate, we have chosen to provide an explanation only at the point of the solution.
>
> **4. Response to Question 3: Instantaneous Regret Bound**
>
> Thank you for your clarification! According to the original definition in the context of online learning, the definition of instantaneous regret should be $\max_{A \in \mathcal{A}} \langle \theta_{A}^*,s_{n, b} \rangle-\langle \theta_{A_{I_{n, b}}}^*,s_{n, b}\rangle$. However, in the specific setting of contextual batched bandit (CBB) that is the focus of this paper, as derived in Appendix line 282 (second inequality), if we denote the upper bound of $|\langle\theta_A^n,s_{n, b}\rangle-\langle \theta_A^*,s_{n, b}\rangle|$ as $U$, then $2U$ serves as an upper bound for instantaneous regret. Thus, in the context of CBB explored in this paper, we are interested in an upper bound for $|\langle \theta_A^n,s_{n, b}\rangle-\langle\theta_A^*,s_{n, b}\rangle|$ and define it as the instantaneous regret bound (as presented in Theorem 2). We intend to incorporate the relationship between our defined instantaneous regret bound and the original instantaneous regret into the final version.

---

> > ### Comment · Reviewer_et1K · 2023-08-17
> >
> > Thanks for your detailed response, which addressed my concern. I would like to raise the rating to 6 and vote for acceptance with the expectation that the authors will incorporate these clarifications into the next version.

---

> > > ### Author Response · Authors · 2023-08-18
> > > **Thank you for your feedback.**
> > >
> > > We sincerely appreciate your valuable feedback and are pleased to have successfully addressed your concern.  We will ensure to include the mentioned clarifications in the final version.

---

### Official Review · Reviewer_mYi5 · 2023-06-28

**Soundness:** 3 good
**Presentation:** 2 fair
**Contribution:** 3 good
**Rating:** 7
**Confidence:** 3

**Summary:**

This paper uses imputation to improve the learning of the reward parameters, even for nonexecuted steps. This results in better variance and empirical benefits over many existing algorithms.

**Strengths:**

1. The empirical benefit seems quite strong over existing algorithms on real datasets.
2. The proposed idea is clear and intuitive.


**Weaknesses:**

1. I found the notation quite difficult to parse, especially at the beginning of page 4. For example, $L_j^n$ was discussed before it was defined.  Moreover, what is $\nu$ in Theorem 2?
2. The improvement seen by imputation is a little tricky to see without knowledge of $C_{\text{imp}}$ in the bound. I find it a little dubious that the amount in which the bias increases by using imputation is hidden by this mysterious constant $C_{\text{imp}}$. The fact that it is monotonically decreasing is not sufficient. I would like to see a more in-depth analysis of this term to understand better whether imputation truly improves our bounds.
3. A similar problem is in Theorem 4. The constant $C_{\text{reg}}$ is presented without a true interpretation. The improvement of the regret seen here over traditional algorithms is unclear. I believe the dependence on $d$ and $T$ achieved here are similar to what is achieved in the literature already. Therefore, understanding the other constants here is important.

**Questions:**

1. I think the paper could be improved by reducing the amount of time discussing the difference between full and partial feedback. I don't believe it adds much to the paper.
2. How does the regret analysis in Theorem 4 compare to recent algorithms tackling Contextual Batched Bandits?

**Limitations:**

The limitations could have been a bit more developed and further extensions could be discussed more.

Overall, I currently vote for acceptance. However, if the authors do not provide a good rebuttal and fix the issues with the limitations, I may have to reduce my score. However, I believe the issues are fixable.

---

> ### Author Rebuttal · Authors · 2023-08-09
>
> # Response to Reviewer mYi5
>
> We are grateful for the time you took to review our paper and grasp the gist of its content.
>
> **1. Response to Weakness 1: Notations**
>
> We have checked and refined the order of symbol introductions and will include a new notation introduction in the final version. In fact, $\nu$ was introduced in line 200. We acknowledge that due to the placement of Table 1, the presentation might have caused confusion.
>
> **2. Response to Weakness 2: Constant $C_{\mathrm{Imp}}$ in Additional Bias**
>
> In fact, we have devoted significant effort to obtain a comprehenscive theoretical analysis of the additional bias, with the details in Appendix B.1. However, for the sake of reader comprehension, we have streamlined it into a constant denoted as $C_{\mathrm{Imp}}$ in the main text.
>
> Now, we summarize the theoretical analysecs for the additional bias:
>
> * During the proof of Thm.2 (Appendix B.1 Line 115-117), it was revealed that in the $n$-th episode, the upper bound of the additional bias can be represented by a function $f_{\mathrm{Imp}}(n):=\sum_{i \in [n-1]}(\sqrt{\eta})^{n-i}CIR_i,$ where $CIR_i $ stands for the cumulative instantaneous regret in the $i$-th episode. Thus, the analysis of the additional bias is transformed into the analysis of the properties of $f_{\mathrm{Imp}}(n)$.
>
> * Lemma B.1 in Appendix B.1 analyzes the convergence and monotonicity of $f_{\mathrm{Imp}}(n)$, revealing that under mild conditions (Remark B.1 of Appendix B.1), $f_{\mathrm{Imp}}(n)$ is monotonically decreasing. This property allows the additional bias to be expressed as $\gamma^{\frac{1}{2}}\eta^{-\frac{1}{2}}f_{\mathrm{Imp}}(n)$, controllable by the imputation rate $\gamma$. This result indicates that the additional bias decreases as the number of episodes increases, which aligns intuitively with the notion that reward imputation leverages an estimated reward model that becomes more accurate with an increasing sample size.
>
> Moreover, we validate theoretical findings with bias, variance, and regret curves in the uploaded PDF (Fig.1 and Fig.2). Our empirical analysis yields these conclusions:
>
> * The additional bias of our approaches diminishes gradually with increasing episodes, and the additional bias compared to the inherent bias (SBUCB's bias) is only a minimal fraction.
>
> * Our reward imputation significantly decreases variance, achieving a nearly 10% reduction compared to the variance of SBUCB.
>
> * The regret of PUIR and SPUIR is smaller compared to SBUCB.
>
> **3. Response to Weakness 3: Cumulative Regret**
>
> The constant $C_{\mathrm{reg}}$ is defined as  $C_{\mathrm{reg}}=\sqrt{\dfrac{\tilde{\sigma}_1^2+\lambda}{\tilde{\sigma}_d^2[1-2\varepsilon/(6+3\varepsilon)]+\lambda}}$ (Appendix, line 269). Here, $\varepsilon$ represents the sketching approximation error, $\tilde{\sigma}_1$ and $\tilde{\sigma}_d$ are the maximum and minimum singular values of context matrix (Appendix B.3, line 227), and $\lambda$ is a regularization parameter. The source of $C{\mathrm{reg}} \geq 1$ is the sketching approximation:
>
> * When sketching is not used for acceleration ($\varepsilon=0$), $C_{\mathrm{reg}}$ reduces to 1, meaning it has no impact on the cumulative regret bound.
>
> * As per the definition of $C_{\mathrm{reg}}$, it decreases with increasing $1/\varepsilon$. The lower bound on the sketch size required for obtaining the cumulative regret bound is given by $c=\mathrm{\Omega}(d\;\mathrm{polylog}\left(d\delta_2^{-1}\right)/\varepsilon^2)$. This implies that for smaller preset sketching approximation errors $\varepsilon$, a larger sketch size $c$ is needed, which in turn reduces the value of $C_{\mathrm{reg}}$ in the cumulative regret bound.
>
> Hence, the parameter $C_{\mathrm{reg}}$ in the cumulative regret bound quantitatively characterizes the trade-off between sketching approximation error and cumulative regret. This serves as a key contribution of Thm.4, differentiating it from the existing batched contextual bandit theoretical analyses.
>
> Due to the existing focus of cumulative regret bounds for batched contextual bandits on the orders with respect to input dimension $d$ and the number of rounds $T$, many constants have been overlooked in their derivation. This perspective might lead to an initial impression that our proposed method does not exhibit a significant improvement in terms of these orders and other constants.
> However, it's important to emphasize that by considering the instantaneous regret bound in Thm.2, we can confidently assert: Within the cumulative regret bound Eq.(10), the cumulative variance terms $\sqrt{10M}\log(T+1)(\sqrt{dT}+dB)$ associated with the proposed approaches have neglected constants that are undoubtedly smaller compared to the corresponding cumulative variance terms of the batched UCB policy without reward imputation. The added experimental results in the uploaded PDF (Fig.1 and Fig.2) validate this observation: it is evident that our PUIR and SPUIR consistently achieve lower cumulative regrets compared to batched UCB policy.
>
> **4. Response to Question 1: Simplified Discussion**
>
> We have condensed the second paragraph of the introduction and streamline the discussion about full and partial feedback in Section 2.
>
> **5. Response to Question 2: Regret**
>
> Compared to recent algorithms addressing contextual batched bandits' regret analysis, our regret analysis quantitatively characterizes the trade-off between sketching approximation error and cumulative regret. This constitutes a significant contribution of Thm.4, setting it apart from existing theoretical analyses of contextual batched bandits (CBBs). The orders of the number of rounds $T$ and the input dimension $d in our regret upper bound, match the lower bound of CBBs. Furthermore, based on the analysis of bias and variance in Thm.2, the cumulative variance term within our regret bound demonstrates reduced magnitude in our regret bound. For detailed information, please refer to the feedback in *Response to Weaknesses 3*.

---

### Official Review · Reviewer_wPev · 2023-07-05

**Soundness:** 3 good
**Presentation:** 2 fair
**Contribution:** 2 fair
**Rating:** 5
**Confidence:** 3

**Summary:**

This paper studies the contextual batched bandit (CBB) problem, where the learner interacts with the environment in an online manner through $N$ episodes. In episode $n$, the learner selects the policy $p_n$,  receives $B$ contexts $s_{n,b}\in \mathbb{R}^d$,  decides the action $A_{I_{n,b}}$ according to $p_n(s_{n,b})$, receives the feedback and updates the policy. The reward is determined by an unknown action-specific reward parameter vector $\theta_{A}^\star$ such that $\mathbb{E}[ R_{i,A} ] = \langle \theta_{A}^\star, s_i  \rangle$ for context $s_i$.

Theorem 1 states the fact that the upper bound of instantaneous regret in the FI-CBB setting is tighter than that in CBB setting, where "FI" means full-information feedback, that is, the learner is able to observe all the losses of both executed and non-executed actions.

Then, this paper proposes reward imputation to estimate the reward feedbacks for the non-executed actions. Roughly speaking, the learner estimates the reward for the non-executed actions by using an imputed reward vector in Line 144. Then, the parameter vector is updated of the imputed reward and the observed reward together (with respect to $\gamma$, the imputation rate) as shown in Equation. (1). To further improve the efficiency, the paper utilizes the sketching technique to reduce the computational cost from $\mathcal{O}(Bd^2)$ to $\mathcal{O}(cd^2)$ where $c$ is the sketch size and assumed to be $d<c<B$ and $dc>B$.

The instantaneous regret bound is stated in Theorem 2. The approximation error bound of using sketching is described in the Theorem 3, and the final regret bound of the reward imputation with sketching is stated in Theorem 4.



**Strengths:**

1. The paper is clearly-written and self-contained. The notations are well-defined prior to the usages.
2. The paper has some nice experiment results for demonstration.

**Weaknesses:**

1. The comparison between the regret bound with/without reward imputation is a little vague: from line 248 to line 250, the authors claim that "in each episode, our policy updating has a smaller variance than the policy without the reward imputation, and incurs a decreasing additional bias under mild conditions, leading to a tighter regret (i.e., smaller policy error) after some number of episodes". I wonder if the authors could be more specific that which algorithm they are comparing to?
2. The assumption "if $\Pi$ and $\widehat{\Pi}$ are SJLT, assuming that $D=\Theta(...)$" is not very clear.  Does this assumption hold usually or with high-probability?
3. Is the batch size $B$ a tunable hyper-parameter? In Remark 3, the authors set $B = \mathcal{O}(\sqrt{T/d})$, while in Protocol 1, the batch size $B$ seems to be fixed prior to the interaction? In addition, the condition that $d<c<B$ and $cd>B$ is not reasoned in details.

[Minor Issues]
1. The problem setting is not well-organized. The interaction protocol is described separately in the introduction and problem formulation sections. I would like to suggest putting all the notations into one section for better readability.


**Questions:**

My questions are raised in the weakness section. I am willing to re-evaluate the score if the questions are answered properly.

**Limitations:**

This work is pure theoretical, and does not have any potential negative societal impact.

---

> ### Author Rebuttal · Authors · 2023-08-09
>
> # Response to Reviewer wPev
>
> Thanks for your comments and suggestions.
>
> **1. Response to Weakness 1: Regret Comparison**
>
> In the discussion of the regret bound theory, we compare our proposed "policy with reward imputation" to a "policy without the reward imputation" that corresponds to the Batch UCB Policy (as described in Algorithm 1 in Appendix A.2). In Eq. (1), if we do not use the imputation term during parameter updating, the policy will degrade to a Batch UCB Policy. The discussion following Theorem 1 (lines 248 to 250) demonstrates that our reward imputation approach tightens the regret bound on the original policy. This is achieved by maintaining a decreasing additional bias while simultaneously reducing variance.
> As empirical validations of the theoretical findings, we have added illustrations of bias, variance, and regret for the proposed algorithm in the newly uploaded PDF (Figure 1 and Figure 2). From the experimental results, it is evident that the proposed reward imputation approach has a positive impact on both reducing additional bias and decreasing variance.
>
> We will incorporate the aforementioned discussion and results into the final version of the paper.
>
> **2. Response to Weakness 2: Assumption on SJLT**
>
> In Theorem 3, the assumption about SJLT is guaranteed to be satisfied. Specifically, $D$ (defined in line 178) represents the number of sub-matrices used during the construction of SJLT. Given the preset values of the error tolerance $\varepsilon$, failure probability $\delta$, and input dimension $d$, the value of $D$ can be determined according to Theorem 3 as follows: $D = \Theta ( \varepsilon^{-1} \log^3 ( d \delta^{-1} ) )$. This theoretical recommendation for $D$ ensures the required properties of SJLT.
>
> **3. Response to Weakness 3: The Batch Size $B$**
>
> The batch size $B$ can be set based on the theoretical value provided in Remark 3. Specifically, the theoretical value of the batch size is $B = C_B \sqrt{T / d}$. In our experiments, given the number of round $T$ and the input dimension $d$, we found that a suitable choice for the constant $C_B$ is approximately 25 for all the datasets (for more details, refer to Table 1 in Appendix C.1).
>
> Additionally, the two conditions for the sketch size $c$, namely $d < c < B$ and $cd > B$, are easily satisfied. Here are the specific settings for the sketch size $c$ for various experiments, as described in Table 1 in Appendix C.1 and Table 2 in Appendix C.2:
>
> * Synthetic data: $B = 1,400$, $d = 40$, $c = 150$.
>
> * Criteo-recent dataset: $B = 1,000$, $d = 50$, $c = 150$.
>
> * Criteo-all dataset: $B = 4,000$, $d = 50$, $c = 150$.
>
> * Commercial product dataset: $B = 1,700$, $d = 50$, $c = 150$.
>
> All of the above experimental settings satisfy the conditions $d < c < B$ and $cd > B$.
>
> **4. Response to Minor Issue**
>
> Thank you for the valuable feedback. We sincerely appreciate your suggestions and will make the necessary changes to improve the organization of the final version. Specifically, we will consolidate the interaction protocol and related notations into one dedicated section titled "Problem Formulation" to enhance readability and clarity for the readers.
>
> We look forward to the possibility of receiving a higher score based on the provided responses. With the limited time remaining, we kindly request you to inform us of any lingering concerns you may have. We are fully committed to addressing any remaining issues to the best of our abilities, and we greatly value the opportunity to refine our work further.

---

> > ### Comment · Reviewer_wPev · 2023-08-17
> > **Thank the authors for their response**
> >
> > Thank the authors for their response. I would like to raise my score to 5, and look forward to the updated version.
> >
> > Specifically, I would like to suggest that:
> > 1. It would be better to emphasize that, the assumption about SJLT is guaranteed to be satisfied.
> > 2. As the interaction protocol is parameterized in B, it would be better to say that, once the B is able to be set by the learner, the regret bound can be further improved.

---

> > > ### Author Response · Authors · 2023-08-18
> > > **Thank you for your feedback.**
> > >
> > > Thank you very much for your feedback and for considering raising your score. We'll highlight the SJLT assumption's guarantee and explain how setting parameter $B$ improves the regret bound in the final version.

---

### Official Review · Reviewer_PXsH · 2023-07-06

**Soundness:** 3 good
**Presentation:** 3 good
**Contribution:** 3 good
**Rating:** 5
**Confidence:** 3

**Summary:**

This paper studies the contextual batched bandits (CBB) setting, where compared to classical online CB, a batch of actions is played and the reward being observed in an episode. Motivated from the benefit of full-information feedback, it tries to perform the reward imputation for unlabelled data, and optimizes the reward parameter by minimizing the combined MSE loss of observed data and imputed data. To further reduce computational complexity, it uses sketching to compress the matrix used in linear regression. Empirically, it shows better performance in both synthetic and real-world experiments.

**Strengths:**

- In the batch bandit literature, the bias-variance trade-off is mostly happened between IPS and reward modeling approaches. This paper focus on another level of bias-variance trade-off within the reward model estimate. It performs reward imputation on unlabelled data then doing regression on top of that. The overall objective contains both the observed fitting loss and the weighted imputed loss, where increasing the weight increases the bias, but potentially decreasing the variance, which gives another level of bias-variance trade-off within the reward model. The idea is interesting.

- The paper is well-written and easy to follow. The method is supported by the theoretical analysis of the instantaneous regret bound.

- To further reduce the complexity, the paper introduces using sketching based on the SJLT, which significantly reduces the time complexity, without losing too much performance.

**Weaknesses:**

- Intuitively, the imputed data in early rounds seem less important (i.e., in terms of accuracy) compared with the imputed data in later rounds, which this requires we basically weigh the imputed data differently, instead of using a fixed $gamma$ for all imputed data. I am not sure how the authors think about this aspect? It would be interesting to see how it works empirically.

- The imputed method seems a way to improve reward models. Fitting a reward model is like a supervised learning problem and there are bunch of ways to reduce variance, like ensemble-based, it would be great to add these baselines, and arguing how the current method helps.

- It seems from Eq 9, the $\gamma$ term could be optimized to minimize the instantaneous regret bound, instead of using a heuristics on this.

- Given there exists synthetic experiments, is it possible to plot the bias, variance, regret for each method? and visualize how the PUIR/SPUIR helps, and how they support the theory.

- I am actually curious to see whether this way of improving reward model also translates to doubly robust based method, where it combines with IPS weighting based approach?



**Questions:**

See weakness part.

---

> ### Author Rebuttal · Authors · 2023-08-09
>
> # Response to Reviewer PXsH
>
> Thank you for your comments and suggestions.
>
> **1. Response to Weakness 1: Setting of Imputation Rate $\gamma$**
>
> Indeed, we have identified the consideration you mentioned regarding the setting of the imputation rate $\gamma$. We have introduced rate-scheduled versions of our PUIR and SPUIR approaches in Section 5 lines 261-266, termed as PUIR-RS and SPUIR-RS.
>
> PUIR-RS and SPUIR-RS eliminate the need for preseting and tunning the imputation rate $\gamma$. Instead of employing a fixed imputation rate, we dynamically adjust the weights of the imputed data by gradually increasing $\gamma$ as the number of episodes increases. This approach helps to mitigate the significant bias that might occur at the initial stages of reward imputation. More specifically, we set $\gamma= X\%$ for episodes ranging from $(X-10)\% \times N$ to $X\% \times N$, where $X \in [10, 100]$.
>
> The experimental results, showcased in Table 2 and Figure 3 within the article, reveal that PUIR-RS and SPUIR-RS perform equally well in terms of reward improvement when compared to PUIR and SPUIR with manually tuned $\gamma$ values. Notably, SPUIR-RS even outperforms SPUIR on the Criteo-all dataset. These experimental findings further validate the analyses we provided concerning the reward imputation bias in Theorem 2.
>
> For additional experimental results regarding rate-scheduled $\gamma$, please refer to Table 3, Table 4, and Figure 2 in Appendix C.3.
>
> **2. Response to Weakness 2: Ensemble-based Baselines**
>
> Thank you for your suggestion. We have incorporated experiments related to "ensemble-based baselines" in the newly uploaded PDF (UCB-AdaBoost and UCB-Bagging in Table 1 and Table 2). In these experiments, "UCB-AdaBoost" refers to utilizing AdaBoost to train the reward model for the batched UCB policy, and "UCB-Bagging" involves employing Bagging to train the reward model for the batched UCB policy.
>
> From the experimental results, it is evident that, in terms of rewards, the improvements brought by ensemble-based baselines are only marginal compared to SBUCB, while introducing additional computational cost. This phenomenon can be primarily attributed to the fact that in the bandit setting, ensemble methods can only enhance the accuracy of reward model estimation based on the limited observed feedback. They do not sufficiently leverage the structural information underlying unobserved reward feedback, thus failing to significantly benefit the exploration-exploitation tradeoff of bandit policy. In contrast, the reward imputation approach proposed in our paper effectively utilizes the structural information of the reward function, thereby enhancing the effectiveness of the exploration-exploitation tradeoff.
>
> **3. Response to Weakness 3: Experimental Curves depicting Bias, Variance, and Regret**
>
> In the newly uploaded PDF, Figure 1 and Figure 2 present experimental curves showcasing bias, variance, and regret. Based on the experimental results, the following conclusions can be drawn:
>
> * The additional bias introduced by our approaches gradually diminishes with increasing episodes, as depicted in Figure 1(a). Moreover, the additional bias in comparison to the inherent bias (bias of SBUCB without reward imputation) is only a marginal fraction, approximately $2\%$, as illustrated in Figure 1(b).
>
> * The proposed reward imputation plays a pivotal role in significantly reducing variance. This achievement is evidenced by a nearly $10\%$ reduction in variance when compared to the variance of SBUCB, as shown in Figure 2(b).
>
> * The regret associated with PUIR and SPUIR is notably smaller compared to the baseline SBUCB without reward imputation. This distinction is illustrated in Figure 2(a).
>
>
> **4. Response to Weakness 4: Optimization Imputation Rate $\gamma$**
>
> We have indeed considered optimizing the imputation rate, denoted as $\gamma$, but encountered significant challenges in doing so. The primary obstacle arises from the fact that, as shown in Equation (9), $\gamma$ is embedded within the inverse matrix in the variance term. This inherent complexity makes a direct optimization of the imputation rate $\gamma$ extremely difficult.
>
> However, we have introduced a parameter-free approach targeting the imputation rate $\gamma$, referred to as the "rate-scheduled approach," which is presented in line 261 of Section 5. This approach has yielded promising results by gradually increasing the value of $\gamma$. Detailed information regarding this approach can be found in the *Response to Weakness 1: Setting of Imputation Rate $\gamma$*.
>
> **5. Response to Weakness 5: IPS Weighting Extension**
>
> Thank you for your valuable insights. As you rightly pointed out, our work distinguishes itself by focusing on a distinct dimension of the bias-variance trade-off within the reward model estimate, achieved through reward imputation on unlabeled data. Our reward imputation approach has the potential to seamlessly integrate with the IPS weighting approach, leading to the development of a novel doubly robust-based methodology. However, such an extension requires some refinements to our approach. For instance, transforming our deterministic policy into a stochastic one.
>
> In existing experimental results, our reward imputation-based bandits exhibited superior performance compared to existing IPS-based baselines (BEXP3-IPW, BLTS-B). In the newly uploaded PDF (PUIR-DR in Table 1 and Table 2), we have added preliminary experimental validation, indicating that our reward imputation-based doubly robust bandit achieved further enhancements in rewards, albeit with an increase in computational cost. The theoretical analysis and efficient algorithm design for the doubly robust bandit based on our reward imputation approach remain a challenging yet promising open research avenue. We look forward to collaborating with domain experts in addressing this intriguing question in the future.

---

> > ### Comment · Reviewer_PXsH · 2023-08-19
> >
> > Thanks for the response! I do think it is a solid work and will keep my score.

---

> > > ### Author Response · Authors · 2023-08-19
> > >
> > > Thank you for your commendation of our work. Your feedback is greatly appreciated.

---

### Author Rebuttal · Authors · 2023-08-09

Thank you for your feedback and valuable suggestions. We have carefully considered and incorporated all the comments and recommendations provided. Additionally, we have uploaded a new PDF file containing supplementary experimental results. We kindly request the reviewers to bring to our attention any remaining concerns, considering the limited time available. We are fully committed to making every effort to address these concerns to the best of our ability.

---

### Decision · Program_Chairs · 2023-09-21

**Decision:**

Accept (poster)

**Comment:**

This paper studies the contextual batched bandits (CBB) setting, where a batch of actions is played together (i.e., the decision has to be made at the beginning at each batch) and the rewards are only observed at the end of this batch.

The main contributions of this paper are: The authors use imputation to improve the learning of the reward parameters, even for nonexecuted steps. This results in better variance and empirical benefits over many existing algorithms. To further reduce computational complexity, the authors propose the usage of sketching to compress the matrix used in linear regression. Empirically, they show better performance in both synthetic and real-world experiments.

There was a strong consensus among the reviewers that this is a good paper and should be accepted. I also fully agree with this opinion, hence I recommend accepting the paper.